# *DOCK3*-Associated Neurodevelopmental Disorder—Clinical Features and Molecular Basis

**DOI:** 10.3390/genes14101940

**Published:** 2023-10-14

**Authors:** Matthew S. Alexander, Milen Velinov

**Affiliations:** 1Department of Pediatrics, Division of Neurology, University of Alabama at Birmingham and Children’s of Alabama, Birmingham, AL 35294, USA; matthewalexander@uabmc.edu; 2UAB Center for Exercise Medicine, University of Alabama at Birmingham, Birmingham, AL 35294, USA; 3Department of Genetics, University of Alabama at Birmingham, Birmingham, AL 35294, USA; 4UAB Civitan International Research Center (CIRC), University of Alabama at Birmingham, Birmingham, AL 35233, USA; 5UAB Center for Neurodegeneration and Experimental Therapeutics (CNET), University of Alabama at Birmingham, Birmingham, AL 35294, USA; 6Department of Pediatrics, Division of Genetics, Rutgers Robert Wood Johnson Medical School, New Brunswick, NJ 08901, USA

**Keywords:** *DOCK3*, neurodevelopmental disorder, genetic disorder, function of *DOCK3*

## Abstract

The protein product of *DOCK3* is highly expressed in neurons and has a role in cell adhesion and neuronal outgrowth through its interaction with the actin cytoskeleton and key cell signaling molecules. The DOCK3 protein is essential for normal cell growth and migration. Biallelic variants in *DOCK3* associated with complete or partial loss of function of the gene were recently reported in six patients with intellectual disability and muscle hypotonia. Only one of the reported patients had congenital malformations outside of the CNS. Further studies are necessary to better determine the prevalence of *DOCK3*-associated neurodevelopmental disorders and the frequency of non-CNS clinical manifestations in these patients. Since deficiency of the DOCK3 protein product is now an established pathway of this neurodevelopmental condition, supplementing the deficient gene product using a gene therapy approach may be an efficient treatment strategy.

## 1. Introduction

Neurodevelopmental disorders (NDDs) are a large and heterogeneous group of conditions affecting cognitive and motor functioning with typical childhood onset [1]. Recent developments in genomic technologies have led to the discovery of a large number of chromosomal rearrangements and single-gene disorders associated with NDDs [2,3,4]. These discoveries have allowed connecting multiple genes into neurodevelopmental networks and disease pathways [5]. Most importantly, identifying new disease-associated genes has allowed for the development of novel treatment strategies using downstream targets of disease-associated pathways or gene therapy approaches. There are now several FDA-approved protocols for the medical treatment of NDDs. Examples of such protocols approved for clinical use are delivering the entire gene or modifying the RNA product [6] in patients with spinal muscular atrophy, and downstream regulation using a small molecule in patients with *PIK3CA* mutations [7]. Finally, identifying the genetic cause for a specific disorder allows for better-informed family planning for the families and makes possible prenatal or pre-implantation diagnoses in future pregnancies.

Mutations in the gene *DOCK3* associated with complete or partial deficiency of gene function have recently been identified and may lead to the future development of specific therapy protocols for these patients.

## 2. History of *DOCK3*-Deficiency

In 2003, Efron et al. reported a 10-year-old male patient with moderate intellectual disability and severe impulse control problems with features of ADHD and conduct disorder [8]. This individual’s routine chromosomal analysis identified an apparently balanced chromosomal inversion in chromosome 3: inv (3) (p14q21). Afterward, 10 additional family members were found to have the same chromosome 3 inversion and similar clinical manifestations. The average IQ of the family members who carried the family inversion was 76.6 while non-carrier family members had an average IQ of 93.6 [8]. Follow-up studies were carried out to characterize the inversion breakpoints [9]. The 3p breakpoint was found to be between exons 19 and 20 of *DOCK3* while the 3q breakpoint was between exons 13 and 14 of gene *SLC9A9*. The authors showed that both genes were expressed in the brain. They concluded that one or both genes may be associated with a genetic form of developmental disability [9].

In 2017, we reported a brother and a sister who presented with the phenotype of intellectual disability and muscle hypotonia [10]. Both siblings had a paternally inherited chromosomal deletion involving *DOCK3* and a maternally inherited loss of function mutation in this gene. Functional studies were not performed. However, *Dock3* knockout mice were previously shown to exhibit gait abnormalities and ataxia, limb weakness, and impairment in learning ability [11]. We, therefore, concluded that the affected siblings’ symptoms recapitulate the phenotype of the *Dock3* knockout mouse model and represent the first reported cases of complete *DOCK3* loss-of-function, causing disease in humans [10].

## 3. Clinical Features of *DOCK3*-Deficiency

The proband in our report [10] was a 12-year-old girl. She is one of seven children of her parents with four sisters and two brothers. The parents are of Ashkenazi and Yemeni Jewish ancestry. Her mother had a difficult labor during her daughter’s birth and needed vacuum extraction. She was born post-mature at 42 weeks. Her neonatal period was uncomplicated. Her development was delayed. She started walking at the age of 5 and had an unsteady gait. At age 12, she needed support while walking. She was nonverbal and not toilet trained. Upon physical examination, her height was 144 cm (15th percentile), her weight was 37 kg (25th percentile), and her head circumference was 54.5 cm (75th percentile). She had a crouched gait and was looking for support while walking. She had a prominent chin, high arched palate, malocclusion, and long fingers. She had decreased deep tendon reflexes in her knees. She did not have spasticity. The rest of her exam was unremarkable.

She had an 11-year-old brother with developmental delays. He was born at 41 weeks without any complications during labor. Sepsis was suspected after birth, but the blood cultures were negative. He started walking at 2.5 years of age. He said his first words at 4 years. His speech at age 11 years was still difficult to understand. He was partially toilet trained. Upon physical examination, his height was 133 cm (8th percentile), his weight was 34 kg (40th percentile), and his head circumference was 51.5 cm (30th percentile). He had an unsteady gait like his sister but did not need support to walk. He had down-slanting palpebral fissures, a long face, and a pointed chin. His deep tendon reflexes were normal and symmetric. There was no spasticity. The rest of his physical exam was unremarkable.

After the initial report, four additional patients from four unrelated families were reported [12,13]. A summary of the clinical features of patients reported to date is shown in Table 1. No common mutations were identified in these reports. All patients had moderate to severe intellectual disability. Two were nonverbal and one had some speech that was later lost. The age of walking ranged from 22 months to 5 years. Variability of clinical severity was observed even within the same family [10]. Four of the reported patients (indicated in Table 1 as P1, P2, P3, and P4) have loss of function mutations, while patients P5 and P6 have missense mutations that were shown to decrease *DOCK3* activity [13]. Multiple congenital anomalies outside the CNS were only present in P4. Except for P3, who had homozygous chromosomal deletion, all patients were diagnosed using whole exome sequencing (WES).

Of interest is the fact that 4 out of the 12 genetic variants in the reported individuals were chromosomal rearrangements involving larger portions of the gene. This brings the question as to whether the *DOCK3* chromosomal region is prone to rearrangements. Accordingly, WES alone may miss some of the patients with chromosomal rearrangements who may be difficult to detect with this technology. Our search of various genetic databases and published reports did not identify additional patients with rearrangements involving *DOCK3* that had similar phenotypes, possibly because biallelic changes are necessary for the development of the *DOCK3*-associated phenotype, while chromosomal rearrangements typically involve only one of the *DOCK3* alleles.

## 4. Etiology and Molecular Pathways

### 4.1. DOCK3 Is Essential for Normal Cell Growth, Proliferation, and Migration

*DOCK3* was first identified as MOCA (modifier of cellular adhesion) or (presenilin-binding protein, PBP) and was shown to be highly expressed in neurons and to function as a guanine nucleotide exchange factor (GEF) by activating the Rho GTPase RAC1 [14]. DOCK3 has been shown to associate with cellular adhesion, specifically by associating with Alzheimer’s disease tangles through the regulation of the accumulation of amyloid precursor protein and β-amyloid plaques [15]. The same study demonstrated that DOCK3 promoted neuronal outgrowth through interactions with N-cadherin and F-actin by promoting cell-to-cell adhesion. There is strong evidence that DOCK factors promote cellar fusion and cell differentiation in multiple cell types, and DOCK3 is not the exception. *DOCK3* transgenic overexpression in mice promoted axonal outgrowth by recruiting the WAVE1 signaling complex to the outer membrane [16]. Neuronal outgrowth is subsequently stimulated by BDNF signaling and RAC1 activation, thereby causing cytoskeletal rearrangement and DOCK3-FYN protein–protein association. RAC1 activation and downstream signaling are one of the key functions of *DOCK3*. Cellular movement and migration require DOCK3 as siRNA DOCK3 in tumorigenic cell lines showed poor motility [17]. RAC1 activity is a key function of DOCK3 in neuronal disease states as found in epileptic patients and models; DOCK3 levels are increased but activated RAC1 (RAC1-GTP) levels are decreased [18]. In Duchenne muscular dystrophy (DMD), a progressive X-linked neuromuscular disorder, *DOCK3* expression is increased; however, the increased expression is observed along with a subsequent decrease in the levels of activated RAC1 due to skeletal myofiber membrane instability [19,20]. Evaluation of *DOCK3* patient pathogenic variants found that most of the damaging alleles were those that disrupted RAC1 binding, which suggests that DOCK3–RAC1 interactions are essential for normal cellular functions in neurons and other important tissues [12,13].

DOCK3 has other key interacting molecular partners that likely play important roles in cellular signaling and development processes within neurons. DOCK3 interacts with ELMO1/2 (engulfment and cell motility 1/2) to regulate axon guidance and cellular polarity after stimulation by Sonic Hedgehog (SHH) [21]. More extensive probing of this complex revealed RhoG as another key interacting member of this complex that also regulates neurite outgrowth as part of a ternary complex [22]. Additional interactions with N-methyl-D-aspartate receptors (NMDARs) play essential cytoprotective roles in preventing cell death in retinal ganglion cells (RGCs) [23]. In non-neuronal cell types, DOCK3 also has key protein–protein interactions. In skeletal muscle myotubes, DOCK3 interacts with SORBS1, a key regulator of glucose and insulin signaling [24]. There are likely additional DOCK3-interacting proteins that may be identified on a tissue-specific and temporal basis that have yet to be identified and may also play significant roles in cellular growth and signaling processes.

### 4.2. DOCK3 Is Comprises Key Evolutionary Conserved Domains Essential for Protein–Protein Interactions

Mammalian DOCK3 is a member of the Dock-B subclass of DOCK proteins along with DOCK4, and the longest human DOCK3 protein isoform is approximately 2030 amino acids (Figure 1A). The human DOCK3 protein consists of a conserved SH3 domain, DHR-1/2 domains, and a PxxP domain. The DHR-2 (DOCK homology region 2) domain of DOCK3 is essential for GEF activity and binds to the WAVE proteins via their DHR-1 domains, thus subsequently activating RAC1 [16]. Comparisons of protein structures among DOCK family members have revealed commonalities that both distinguish DOCK proteins from other GEFs and define the mechanism by which a DOCK catalytic DHR2 domain elicits nucleotide dissociation from small RhoGTPase (especially, RAC1) [25]. The SH3 domain in DOCK proteins is likely responsible for the signal transduction of tyrosine phosphorylation signals to the actin cytoskeleton in regulating cell motility [25,26,27,28]. Subsequently, the SH3 domain also interacts with the PxxP domain within each of the DOCKs (DOCK1-5) to initiate actin signals [29,30]. As more information emerges on the structure of DOCK3, DOCK proteins, and DOCK3 protein interactions, increased knowledge of DOCK3’s functional role in different cell types and the consequences of loss-of-function variants will be better understood.

A full protein structure of mammalian DOCK3 does not exist; however, several AI-predicted structures for DOCK3 exist including one for human DOCK3 from AlphaFold (Figure 1B) [31]. Insights from the published cryo-EM structure of the DOCK2-ELMO2 protein–protein complex may yield clues into commonalities for protein binding mechanisms between DOCK3 and ELMO binding [32]. DOCK2–ELMO1 interactions mutually relieve their autoinhibition for the activation of RAC1 for lymphocyte chemotaxis, which might explain similar DOCK3-ELMO1/2 regulation of cellular migration [33]. DOCK3 binding to ELMO1/2 likely similarly regulates GEF activity and promotes subsequent cellular growth. While DOCK3 is conserved in vertebrate species, in Drosophila, a DOCK3/4 counterpart is the *sponge* (*spg*)/CG31048 gene, which also interacts with ELMO and regulates RAC1 activity [34]. Knockdown of *sponge* in Drosophila eye imaginal discs induced abnormal eye morphology in adult flies and reduced ERK signaling [35]. Follow-up Drosophila studies demonstrated a critical role for *sponge* in air sac promordium development and tracheal cell viability that was mediated by the ERK signaling pathway. Further study of DOCK3 domains in other species and cell models may yield additional clues into the function of DOCK3 in a cellular and tissue signaling context.

Evaluation of the functional role(s) of DOCK4 may also yield clues into novel approaches in corrective therapies for *DOCK3* patients. An interesting finding from a recent evaluation of *Dock4*-deficient (*Dock4* KO) mice showed that overexpression of active RAC1 viral vectors could partially rescue neuronal functional tests [36]. Although it is not desirable to fully activate RAC1 in every tissue due to its role as a known cancer-driving oncogenic factor [37], temporal and tissue-specific modulation of active RAC1 may be beneficial for rescuing some of the DOCK3-affected pathways that are altered in patients. Indeed, some pharmacological modulators of RAC1 activity are being explored for cognitive disorders and may be worth pursuing in *Dock3*-deficient mice and cell lines [38]. If RAC1 modulators were both tissue-specific and reversible, there may be benefits towards exploring the effects of RAC1 modulation in *DOCK3*-deficiency in animal models with the end goal of patient trials should efficacy and safety be established.

### 4.3. Dock3-Deficient Mice Have Significant Developmental and Regenerative Defects

*Dock3* knockout (*Dock3* KO) mice were generated and showed progressive movement defects and accumulation of autophagic vacuoles accumulating in the spinal cords of aged *Dock3* KO mice [11]. *Dock3* KO mice develop clasping pathologies and abnormal aggregates of neurofilament protein along with the disorganization of the axonal cytoskeleton. Electron microscopy of *Dock3* KO mice revealed impaired axonal transport of and a general accumulation of polyubiquitinated proteins, a hallmark of neurodegenerative disorders [11,39]. Genetic ablation of *Dock3* on the dystrophin-deficient (*mdx*) background further exacerbates dystrophic pathologies including increasing muscle fibrosis and loss of muscle physiological force [20]. Interestingly, haploinsufficiency of *Dock3* (*Dock3* +/−) on the *mdx* background restores the DOCK3 protein to normal levels and partially improves dystrophic muscle pathologies and histology. Conditional disruption of *Dock3* in skeletal muscle (*Dock3* mKO) results in mild impairment of muscle pathology and histology including impairment of glucose processing [20]. Additional use of conditional *Dock3* knockout mice to evaluate DOCK3′s functional role in other cell lineages may yield clues in other tissue and cell types.

### 4.4. RAC1-Affected Pathways Affected by DOCK3 Disruption

As previously mentioned, genetic disruption of *DOCK3* in patients results in decreased RAC1 activation [13]. This *DOCK3*-deficiency and decrease in global RAC1 activity likely has profound affects throughout several tissues as RAC1 is a key regulator of cellular migration, fusion, differentiation, proliferation, and viability as a GTPase effector protein [21,40]. RAC1 is a dynamic molecule that can have profoundly different effects in a tissue-specific and cellular context. Patients with ultra-rare *RAC1* missense mutations have reported symptoms consisting of developmental delay, and macrocephaly, although some reported instances of microcephaly depended on whether the *RAC1* mutation was a dominant negative activating or inactivating mutation [22,41]. Interestingly, those same *RAC1* variants were able to be effectively modeled in zebrafish through the use of mRNA overexpression and the quantification of zebrafish larvae head circumference, highlighting the utility of animal models in dissecting *RAC1*-specific mutational consequences.

Conversely, complete loss of *Rac1* expression in *Rac1* knockout (*Rac1* KO) mice is embryonic lethal due defects in the formation of the embryonic germ layers resulting from the cellular disruption of lamellipodia formation, cell adhesion, and cell migration processes [23,42]. Further elucidation of RAC1’s functional roles occurred with the generation of Rac1 conditional knockout mice (*Rac1* cKO) which demonstrated the tissue-specific requirements of RAC1. For example, cardiomyocyte-specific *Rac1* knockout mice (c-*Rac1*) demonstrated an essential requirement for RAC1 in cardiac hypertrophy [24,43]. This study revealed that *Rac1* expression in cardiomyocytes was essential for NADPH oxidase activity and myocardial oxidative stress in response to angiotensin II treatment. This study along with others was one of the first indicators of tissue-specific roles for RAC1 and highlighted the utility of the *Rac1* cKO mouse.

Follow-up work from the laboratory of Lykke Sylow and colleagues further expanded upon this model through the conditional ablation of *Rac1* in skeletal myofibers using an inducible muscle-specific promoter (*Rac1* mKO) [25,44]. *Rac1* mKO mice given a high-fat diet (HFD) experienced insulin resistance in their skeletal muscles and in vivo insulin-stimulated glucose uptake was reduced in the triceps, soleus, and gastrocnemius muscles. Interestingly, whole-body glucose uptake was unaffected when assessed using a 2-deoxy-glucose uptake challenge, as was AKT activation. These results suggest that *Rac1* genetic loss is detrimental to insulin-stimulated muscle glucose uptake independent of skeletal muscle AKT signaling pathway activation. Interestingly, studies of these *Rac1* mKO mice also showed that exercise enhanced whole-body insulin sensitivity by 40% in WT mice and rescued insulin intolerance in *Rac1* mKO mice by improving whole-body insulin sensitivity by 230% [26,45]. What was equally important from these studies was that exercise improved insulin muscle sensitivity in both WT and *Rac1* mKO mouse muscles, suggesting that RAC1 dysfunction may be dispensable for the correction of muscle insensitivity. More recent work from the Sylow group has identified RhoGDIα phosphorylation as a critical step towards skeletal muscle GLUT4 translocation and RAC1 activation [27,46]. When bound to RAC1, RhoGDIα inhibits RAC1 activation, although it is unclear if DOCK proteins interact directly or indirectly with this complex.

Additional studies by other groups have focused on the dynamic regulation of RAC1 to Rho signaling in the context of other cell populations that reside within the skeletal muscle. In quiescence muscle satellite cells (MuSCs), activated RAC1 inhibits activated RhoA, which is inversely switched in early activated MuSCs [28,47]. These activities control cellular projections and filopodia formation within the MuSC population via F/G-actin polymerization. In an adeno-associated viral vector (AAV)-mediated motor neuron *Rac1* knockout mouse model, these mice developed abnormal dendritic spine morphology associated with hyperexcitability disorder, increased mature, mushroom dendritic spines, and an increased level in overall spine length and spine head size [29,48]. Interestingly, after a spinal cord injury (SCI), three-weeks post injury, it was observed that there was significant restoration of rate-dependent depression (RDD) and reduced H-reflex excitability in these *Rac1* motor neuron knockout mice. More recently, the topic of RAC1 regulation has been of interest as a means for understanding the dynamics of its regulation. In skeletal muscle, a muscle-enriched microRNA signaling pathway “myomiRs” (miR-1/206/133) has been shown to regulate DOK7-CRK-RAC1, which is critical for the stabilization and anchoring of postsynaptic acetylcholine receptor AChRs during NMJ development and maintenance [30,49]. The authors concluded that the failure to properly modulate RAC1 activity severely compromises NMJ function, causing respiratory failure in neonates and neuromuscular symptoms in adult mice. These findings highlight the dynamic nature of RAC1 expression and the regulation of RAC1 signaling in different muscle cell populations. Nevertheless, the dynamics of RAC1 activation and its protein–protein binding partners in skeletal muscle and neuronal tissues remains to be fully elucidated.

The role of RAC1 activation and DOCK proteins also appears to be that of a tissue- and cell-specific nature. RhoG was shown to activate RAC1 via DOCK1 and ELMO1 to affect mammalian cell shape and migration [31,32,50]. Follow-up studies demonstrated that activation of RAC1 by RhoG is required for normal lamellipodia formation at the leading edge during cellular migration [33,51]. In contrast, this study also demonstrated that the interaction of DOCK1 with CRK was dispensable for the activation of RAC1 and the promotion of cell migration by RhoG. The generation of additional *Dock1* and *Dock5* knockout mice revealed essential, non-overlapping functions for each factor in myoblast fusion and muscle differentiation [34,52]. Similar work from our group not only demonstrated that *Dock3* knockout mice also had impaired muscle myoblast differentiation and regeneration following injury through downregulation of the fusogen Myomixer (*Mymx*) but also failed RAC1 activation [19,20]. Interestingly, recent findings from the work of Jean-François Côté and colleagues demonstrated that dysferlinopathy (limb girdle muscular dystrophy type 2B/R2) muscle pathologies can be partially mitigated by manipulating ELMO2 conformational regulation [35,53]. Whereas the authors demonstrated that both ELMO1 and ELMO2 muscle expression were required for myoblast fusion, manipulation of key regulatory domains could affect muscle outcomes. ELMO proteins are normally in a closed conformation and the DOCK1 DHR-2 domain is blocked by the Ras-binding domain (RBD) domain of ELMO, subsequently preventing RAC1 activation and the binding of interacting proteins to ELMO proteins. Upon activation of the ELMO-DOCK1 complex, protein–protein binding sites open up to allow for RAC1 activation. It remains unclear if a similar process occurs in DOCK3 or other DOCK proteins that bind to both ELMO proteins and RAC1 regulators; however, these studies have laid the foundation for future work in this field. Newer, exciting drug screens for small molecule compounds that modulate DOCK3 conformational changes affecting DOCK3 and ELMO1 interactions have recently been published and could be explored for treating axonal injury and neurodegenerative diseases [36,54].

### 4.5. Disease Phenotypes Associated with Deficient Functioning of Genes Interacting with DOCK3

The *RAC1* gene is activated by *DOCK3*. It encodes an RHO GTPase involved in the modulation of the cytoskeleton, which plays a role in multiple cellular functions, including phagocytosis, mesenchymal-like migration, neuronal polarization, axonal growth, and the differentiation of multiple cell types. *RAC1* is also involved in cellular growth and cell-cycle regulation [41]. De novo, heterozygous variants in *RAC1* are associated with a distinct NDD that includes dysmorphic features, moderate/severe intellectual disability, seizures, hypotonia, CNS anomalies including cerebellar dysplasia, hypoplasia of the corpus callosum, enlarged ventricles, mega cisterna magna, a thin brainstem, white matter abnormalities, and polymicrogyria [41,55]. In a study, 14 cases with de novo, heterozygous *RAC1* mutations were reported [41,55]. Of the six cases that had functional studies, four were shown to be gain-of-function type and two had a dominant-negative effect. While there are significant similarities in the manifestations associated with *DOCK3* and *RAC1* variants, there are also differences. For instance, the RAC1 group showed brain malformations not described in association with *DOCK3* deficiency. In summary, the relationship between *DOCK3* disfunction and RAC1 is not fully understood and needs further study.

Another disease phenotype is associated with the gene *ELMO2* which interacts with *DOCK3*. The *ELMO2*-associated condition includes vascular malformations with recurrent and sometimes life-threatening bleeding in different organs. This phenotype was reported in eight individuals from five families and was associated with biallelic loss of function variants in *ELMO2* and autosomal recessive inheritance [56]. The relationship between *DOCK3* deficiency and *ELMO2* functioning also remains to be clarified in future studies. The clinical consequences of *DOCK3* deficiency seem to be very different from those associated with *ELMO2* deficiency.

Finally, the DOCK3 protein was initially characterized as a presenilin-binding protein [14]. Variants in presenilin 1 and 2 are associated with autosomal dominant Alzheimer’s disease and dilated cardiomyopathy [57].

## 5. Future Research

The future of *DOCK3*-related research is likely tied to that of other DOCK factors in determining the function of *DOCK3* on a tissue-specific basis. Research into *DOCK3* and its other DOCK-B family member *DOCK4* may yield clues into its function in other tissue types in development and disease [58]. The use of conditional knockout animals and CRISPR-genomic editing approaches will likely yield clues into its function in a cell-type and temporal-specific manner. DOCK3 is an important factor in many tissues outside of neurons and improved DOCK3 antibody and protein detection strategies will yield better insight into the localization of DOCK3 within mammalian healthy and disease tissues. Probing of DOCK3 expression in cancer and other atlas databases (e.g., The Cancer Genome Atlas—TCGA) may yield insights into DOCK3 dysregulation in disease contexts.

Given the large size of the human *DOCK3* gene (53 exons; 6 kb open reading frame), traditional overexpression restorative gene therapies are unlikely due to viral packaging limitations. However, one can look towards the neuromuscular field for insight into strategies for genetic correction of partially functional DOCK3 proteins using exon-skipping phosphorodiamidate morpholino oligomers (PMOs). In Duchenne muscular dystrophy (DMD), exon-skipping PMOs have been FDA-approved and lead to long-term improvements in DMD patient functional outcomes [59,60,61]. One can envision that exon-skipping PMOs might be a reasonable strategy towards making *DOCK3* patient variants amenable by restoring the *DOCK3* reading frame in one or both pathogenic alleles. Another strategy that may be promising would be a split intein adeno-associated viral (AAV) vector approach to allow for the recombination of the full-length *DOCK3* open reading frame within the cell or tissue of interest. These approaches rely on the reconstitution of separate AAV viral vectors into a single large construct, often using a base-editor approach within the tissue of interest within an organism [62,63]. This approach has shown promise in correcting liver metabolic disease in the *Pah^enu2^* mouse model [60]. With improvements in CRISPR base editing, gene therapy delivery systems, and AAV expression vectors, one can envision testing corrective gene therapy approaches in *Dock3*-deficient mice as a viable strategy for eventual *DOCK3* patient testing.

An additional hurdle in the development of an efficient protocol for gene therapy is the limited efficiency of the viral vectors to cross the blood–brain barrier (BBB). We have recently shown that AAV-based vectors cross the BBB with very high efficiency in a mouse model of fragile X syndrome [64]. While the vector used by us provides such high efficiency only for selected mouse strains, more recently developed AAV vectors show promise for high BBB crossing efficiency in humans [65]. When vectors with high efficiency are used, another potential problem may be oversupply of the gene product. Using highly efficient AAV vectors, we observed a potential decrease in the efficiency to correct the disease phenotype due to oversupply of the protein product of the *FMR1* gene, FMRP [64]. Following these concerns, it is important to remember that *DOCK3* activates RAC1, and at least some of the *RAC1* pathogenic variants were associated with gain of function [41,55].

Alternative to gene therapy, small-molecule agents may be used to correct *DOCK3* deficiency. In a recent study, the authors screened 462,169 low-molecular-weight compounds and identified compounds that stimulate the interaction between DOCK3 and Elmo1, resulting in neurite outgrowth in vitro. Some of these compounds stimulated neuroprotection and axon regeneration in a mouse model of optic nerve injury. These findings may be the basis of the development of new treatments for *DOCK3* deficiency using small-molecule compounds that may be more efficient in crossing the BBB [54].

From a clinical point of view, one important question regarding phenotype–genotype correlations that may need further clarification is the association of *DOCK3* mutations with congenital anomalies. Multiple congenital anomalies outside the CNS were present only in one out of the six of the reported patients to date (P4). In addition, in most reported patients, diagnosis was achieved with WES. Since all reported patients to date have muscle hypotonia and an abnormal gait, the gene may be potentially included in a more targeted muscle weakness test panel. Questions remain as to how to best treat *DOCK3* loss-of-function patients as these patients suffer from intellectual disability and muscle hypotonia from an early age, with them often having difficulty obtaining an official genetic diagnosis without genomic testing. Documentation as to the region of DOCK3 pathogenic mutation(s) would likely provide insight as to the degree of damage influencing protein function. Early assessments of biallelic *DOCK3* pathogenic variant families appear to indicate that larger, multi-exon deletions in the N-terminus of the *DOCK3* gene affect patient outcomes more severely than those with C-terminal *DOCK3* variants or deletions [13]. As more patients are identified with *DOCK3* pathogenic variants and other *DOCK* gene loss-of-function variants, strategies for the treatment of these disorders need to be addressed [66,67]. Gene therapy approaches may need to be refined as traditional overexpression approaches may need to be optimized given the large size of *DOCK3* gene open reading frames.

## Figures and Tables

**Figure 1 genes-14-01940-f001:**
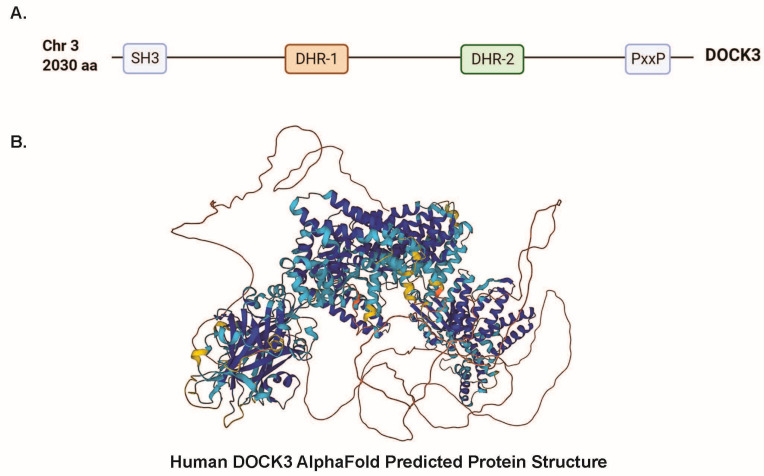
DOCK3 is comprised of key evolutionary conserved domains essential for protein-protein interactions. (**A**) Conserved protein domains for the largest isoform of the human DOCK3 protein (2030 amino acids, aa). These domains consist of an SH3 domain, a DHR-1 and DHR-2 domain, and a PxxP domain. (**B**) The human DOCK3 AlphaFold Predicted Protein Structure for DOCK3 is shown here. AlphaFold AF-Q8IZD9-F1 model is displayed here.

**Table 1 genes-14-01940-t001:** Summary of clinical features of patients with DOCK3 mutations.

Patient	*DOCK3* Variant	Reference	Sex	Age at Evaluation	Birth History	Family History	Developmental Milestones	Growth (%)	Dysmorphic Features	Congenital Anomalies	Studies Prior to Diagnosis
P1	del3:50789040- 51247265/c.382C>T	[10]	F	12 years	Born at 42 weeks gestation	Similarly affected sibling	Severe Developmental Delay, walked at 5 years, unstable crouched, ataxic gait, non- verbal, and not toilet trained at 12y	WT = 25; Ht = 15; HC = 75	Prominent chin, high arched palate, malocclusion, long fingers	None	Normal metabolic screen, EEG, BEAR, brain MRI
P2	del3:50789040- 51247265/c.382C>T	[10]	M	11 years	Born at 41 weeks gestation	Similarly affected sibling	Walked at 2.5 years, first word at 4 years, single words at TOE, unstable, ataxic gait	Wt = 40; Ht = 8; HC = 30	Pointed chin, down slanting palpebral fissures, long face	None	None
P3	homozygous del. 3:51,062,402–51,232,768	[12]	M	28 months	NR	Parents are first cousins	Started sitting at 14 months, walked at 22 months, unsteady gait, few specific words at TOE, Bayley score <50 (at TOE)	Wt = 4; Ht = 5; HC = 14	Epicanthal folds, up-turned nasal tip, prominent cheeks	None	Brain MRI-dysmorphic Corpus Callosum, ECHO-normal
P4	c.1038-2A>G:IVS12- 2A>G/c.3107_3110delACTT	[13]	M	5 years	Born at 37 weeks gestation	Unremarkable	Started walking at 36 months, 5-10 single words at 5	Wt ≥ 99; Ht = 66; HC = 85	Broad forehead, deep set, hooded eyes	TE fistula with esophageal atresia, vertebral anomalies, rib anomalies, single kidney	Negative microarray, brain MRI- shallow sulci, hypoplastic white matter, spine MRI-syrinx, abnormal EEG
P5	c.1175G>A/c.3887A>G	[13]	M	5.5 years	Full term	Unremarkable	Was able to sit at 30 months, walked at 48 months, non-verbal, autism, unprovoked laughter, hypotonia	Wt = 50; Ht = 25; HC = 7	Brachicephaly, plagiocephaly, prominent philtrum	Phimosis	Brain MRI-diminished white matter, hypoplastic CC, negative macroarray, UBE3A, MECP2, meth-Angelman
P6	c.5020A>T/5020A>T	[13]	F	3 years	born at 35 weeks gestation, feeding difficulties	NR	Walked at 18 months, and said the first word at 15 months, but then lost her speech, autism	Wt ≥ 99; Ht ≥ 99; HC ≥ 99	Macrocephaly, frontal bossing	Spina bifida	Brain MRI-resolved Chiari malformation, negative CMA, PTEN, FXS

## Data Availability

No new data were created or analyzed in this study. Data sharing is not applicable to this article.

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
