# Peer review of "DOCK3-Associated Neurodevelopmental Disorder—Clinical Features and Molecular Basis"

_genes, 2023, doi:10.3390/genes14101940_

Round 1

Reviewer 1 Report

The authors propose on overview about DOCK3 loss of function and the related phenotype. The topic is interesting beacuse of the rarity of the disorder. The manuscript is well written, although a little bit redundant in the section resembling the interactions of the protein.

A major issue to include in the discussion is: when does DOCK3 play a role in the brain development? If its action is important in the prenatal age, what can we expect from a gene therapy starting after the molecular diagnosis?

It would be worth of interest a brief discussion of other disorders, if present, related to RAC1 pathway

Finally, I think it’s important to underline that the molecular diagnosis of this disorder can offer some criticalities. An approach by exome sequencing could miss some of the molecular defects, because WES (and some panels as well) can fail in the detection of large intragenic deletions, which are frequent in this condition

Some minor observations:

Both siblings had A paternally inherited chro- 40 mosomal deletion and A maternally inherited loss of function mutation

You can’t indicate the centile with the “ %”

Author Response

Thank you for your notes, here are our answers:

Reviewer 1

The authors propose on overview about DOCK3 loss of function and the related phenotype. The topic is interesting beacuse of the rarity of the disorder. The manuscript is well written, although a little bit redundant in the section resembling the interactions of the protein.

A major issue to include in the discussion is: when does DOCK3 play a role in brain development? If its action is important in the prenatal age, what can we expect from gene therapy starting after the molecular diagnosis?

Answer: we have reported the known roles of DOCK3 in brain development. There are some aspects of this gene’s function that are not fully investigated and need further studies.

It would be worth of interest a brief discussion of other disorders, if present, related to the RAC1 pathway

Answer: We have included an additional section discussing RAC1 and ELMO2-associated disorders

Finally, I think it’s important to underline that the molecular diagnosis of this disorder can offer some criticalities. An approach by exome sequencing could miss some of the molecular defects, because WES (and some panels as well) can fail in the detection of large intragenic deletions, which are frequent in this condition

Answer: we have addressed this issue in the section “Clinical Features of DOCK3-deficiency”

Some minor observations:

Both siblings had A paternally inherited chro- 40 mosomal deletion and A maternally inherited loss of function mutation

Answer: addressed

You can’t indicate the centile with the “ %”

Answer: addressed

Reviewer 2 Report

none

In this review manuscript, Alexander and Velinov describe the potential role of DOCK3 in intellectual disability and muscle weakness. The review is well-organized and easy to read.

I have several suggestions for the authors to expand the review by incorporating 1) information about the size of the inversion on chr. 3, 2) its size, 3) how many genes are present in the rearranged chromosomal segment, and 4) any additional information from both human and animal studies that are available, to demonstrate that DOCK3 is indeed the responsible gene and not private to the particular family they have reported in the review.

Additionally, it would be helpful to report whether any existing GWAS of developmental intellectual disability (DID) have reported any variants in the DOCK3 gene. It will be further advantageous to report if existing studies have assessed the differential expression of DOCK3 protein or mRNA levels in human postmortem brain tissues. All of these studies, in conjunction with the family genetic studies, in more than one family, will increase the reliability of DOCK3 as a potential gene in DID. 

Author Response

Thank you for your notes, here are our answers:

Reviewer 2

Comments and Suggestions for Authors

none

Comments on the Quality of English Language

In this review manuscript, Alexander and Velinov describe the potential role of DOCK3 in intellectual disability and muscle weakness. The review is well-organized and easy to read.

I have several suggestions for the authors to expand the review by incorporating 1) information about the size of the inversion on chr. 3, 2) its size, 3) how many genes are present in the rearranged chromosomal segment, and 4) any additional information from both human and animal studies that are available, to demonstrate that DOCK3 is indeed the responsible gene and not private to the particular family they have reported in the review.

Answer: The reported inversion is balanced (no missing chromosomal material). The only affected genes are DOCK3 and SLC9A9 since the chromosomal breakpoints affect these two genes. There are 6 reported cases of biallelic mutations in DOCK3 that have similar phenotypes (see Table 1 in the manuscript). These cases resemble the available mouse models referred to in the manuscript. This information is in our view sufficient to support the association of DOCK3 mutation with the described neurodevelopmental disorder.

Additionally, it would be helpful to report whether any existing GWAS of developmental intellectual disability (DID) have reported any variants in the DOCK3 gene. It will be further advantageous to report if existing studies have assessed the differential expression of DOCK3 protein or mRNA levels in human postmortem brain tissues. All of these studies, in conjunction with the family genetic studies, in more than one family, will increase the reliability of DOCK3 as a potential gene in DID. 

Answer: There are 6 reported patients from 5 families that have biallelic mutations in DOCK3 and similar phenotypes. This information is in our view sufficient to support the association of DOCK3 mutation with the described neurodevelopmental disorder.

Round 2

Reviewer 2 Report

Please confirm that you have received my recommendation to accept this manuscript for publication. The system is acting up, and I cannot say whether you have received my recommendation or not.